# Electro-Stimulation System with Artificial-Intelligence-Based Auricular-Triggered Algorithm to Support Facial Movements in Peripheral Facial Palsy: A Simulation Pilot Study

**DOI:** 10.3390/diagnostics14192158

**Published:** 2024-09-28

**Authors:** Katharina Steiner, Marius Arnz, Gerd Fabian Volk, Orlando Guntinas-Lichius

**Affiliations:** 1Department of Medical Engineering, University of Applied Sciences Upper Austria, 4020 Linz, Austria; katharina.steiner@medel.com; 2MED-EL Elektromedizinische Geräte GmbH, 6020 Innsbruck, Austria; 3Department of Otorhinolaryngology, Jena University Hospital, 07747 Jena, Germany; mariustaro.arnz@med.uni-jena.de (M.A.); fabian.volk@med.uni-jena.de (G.F.V.); 4Facial-Nerve-Center, Jena University Hospital, 07747 Jena, Germany; 5Center for Rare Diseases Jena University Hospital, 07747 Jena, Germany

**Keywords:** auricular muscles, facial muscles, human, facial palsy, synkinesis, functional electrostimulation, closed-loop system, machine learning, neural network

## Abstract

Background: Facial palsy causes severe functional disorders and impairs quality of life. Disturbing challenges for patients with acute facial palsy, but also with those with chronic facial palsy with synkinesis, are the loss of the ability to smile and insufficient eyelid closure. A potential treatment for these conditions could be a closed-loop electro-stimulation system that stimulates the facial muscles on the paretic side as needed to elicit eye closure, eye blink and smile in a manner similar to the healthy side. Methods: This study focuses on the development and evaluation of such a system. An artificial intelligence (AI)-based auricular-triggered algorithm is used to classify the intended facial movements. This classification is based on surface electromyography (EMG) recordings of the extrinsic auricular muscles, specifically the anterior, superior, and posterior auricular muscle on the paretic side. The system then delivers targeted surface electrical stimulation to contract the appropriate facial muscles. Results: The evaluation of the system was conducted with 17 patients with facial synkinesis, who performed various facial movements according to a paradigm video. The system’s performance was evaluated through a simulation, using previously captured data as the inputs. The performance was evaluated by means of the median macro F1-score, which was calculated based on the stimulation signal (output of the system) and the actual movements the patients performed. Conclusions: This study showed that such a system, using an AI-based auricular-triggered algorithm, can support with a median macro F1-score of 0.602 for the facial movements on the synkinetic side in patients with unilateral chronic facial palsy with synkinesis.

## 1. Introduction

Peripheral facial palsy is a medical condition that affects the facial nerve, leading to weakness or paralysis on one side of the face. This nerve is crucial for controlling facial muscles. The facial asymmetry and the loss of muscle control caused by facial palsy can significantly impact the quality of life [1]. A notable complication of facial palsy in the chronic stage is synkinesis, an involuntary facial movement triggered by an intended one, resulting in muscles being incorrectly controlled [2,3]. Despite available treatments like specialized facial therapy (physiotherapy, ergotherapy, logopedics), medical treatments (botulinum toxin injections) and surgery, patients face significant challenges. The most important ones are the loss of the ability to smile due to dysfunction of the zygomaticus major muscle (ZM) and insufficient eyelid closure due to dysfunction of the orbicularis oculi muscle (OOM) [2,4,5,6,7]. These issues have physical consequences, like eye damage due to poor lid closure and problems during eating and drinking, but also social and psychological effects, underscoring the need for further treatment options [8,9].

Functional electrostimulation offers a promising solution to both challenges by stimulating the facial muscles on the paretic side as needed to enable facial movements, similar to those on the healthy side [10,11,12]. Previous attempts to address this issue, such as Frigerio et al.’s proposal of using infrared-based blink-detecting glasses for facial pacing [13], were not further developed. More recently, Cervera-Negueruela et al. developed a wearable device, which detects the eye blink on the healthy side through active electromyography (EMG) electrodes and electrically stimulates the eye on the paretic side through a stimulator with surface stimulation electrodes, based on a single EMG feature [14]. Ilves et al. [15] and McDonnall et al. [16] demonstrated the effective stimulation of the OOM using the OOM of the healthy side as a trigger, showing promising results with this method. Nevertheless, the already existing approaches use the signals from the healthy side as a trigger and only restore blinking.

Identifying a trigger location on the paretic side can significantly reduce the complexity of the potential stimulation device and preserve the healthy side of the patient. Additionally, the restoration of blinking, eye closure, and smiling would address the most important challenges for patients with facial palsy and extend the scope of the already existing studies. Therefore, prior research focused on the identification of facial movements by analyzing the intramuscular EMG signals from the ZM and OOM on the paretic side [17]. However, this approach is only applicable to patients with synkinetic reinnervation, where facial muscles are affected by synkinesis, and is not suitable for patients with denervated muscles, because denervated muscles lack the necessary connections to generate detectable EMG signals. A machine learning (ML) algorithm was used, classifying intended facial movements based on EMG signals with an average accuracy of 92%. Hochreiter et al. analyzed the surface EMG of the auricular muscles in order to investigate its use as a potential trigger for such a stimulation device [18]. The auricular muscles, like the facial muscles, are innervated by the facial nerve and are affected by synkinetic reinnervation, as seen in the mimic muscles, making them a good trigger source [19]. The auricular muscles are vestigial structures without a function in human beings [19]. Hence, manipulation by recording on the surface (or, in the future, by an implantable device) cannot lead to a disturbance of normal function. Hochreiter et al. investigated the machine-learning-based detection of eye closure and smile using the surface EMG of the auricular muscles on the paretic side [18]. They demonstrated the feasibility of using auricular EMG signals, which are effected by synkinetic reinnervation, for detecting facial movements.

Although the auricular muscles have been shown to activate during facial movements [20,21], and their EMG signals have been used to control devices like wheelchairs and prostheses [22,23,24], their potential as a robust trigger for electro-stimulation devices in facial palsy treatment has not been fully explored. Hochreiter et al.’s work demonstrated the feasibility of using auricular EMG signals to detect facial movements like eye closure and smiling, but a more comprehensive system that can classify multiple facial movements in real time is yet to be developed.

The present study addresses these limitations by developing and evaluating a closed-loop electro-stimulation system for patients with synkinetic chronic peripheral facial palsy. Unlike prior systems that relied on the healthy side or complex intramuscular EMG signals, this approach leverages the extrinsic auricular muscles of the paretic side as a reliable trigger source. These muscles are innervated by the facial nerve, involved in synkinesis, and easily accessible. The system measures the extrinsic auricular EMG signals, classifies the intended facial movement, and delivers targeted surface stimulation to the appropriate facial muscle in real time. Compared to the previous systems, which use simple algorithms and restore only blinking, a key component in the present system involves an AI-based auricular-triggered algorithm to identify the facial movements of eye closure, eye blink, and smile from the auricular EMG data. This closed-loop design, which enables real-time classification and stimulation, represents a significant advancement over previous methods by improving the system’s applicability and preserving the natural function of the healthy side.

## 2. Material and Methods

This pilot study was conducted in the Facial Nerve Center Jena at Jena University Hospital, Jena, Germany. It was approved by the local ethics committee from University Hospital Jena (2023-3169-BO). Written informed consent was obtained from all patients.

### 2.1. Study Population

Twenty patients (12 female, 49 ± 13 years; 8 male, 58 ± 15 years) diagnosed with unilateral peripheral chronic (facial palsy duration: female 27 ± 15 months, male 77 ± 105 months) facial palsy with synkinesis (Sunnybrook [25] synkinesis scores: females 9 ± 2, males 9 ± 3) were recruited. All patients were selected from the Facial Nerve Center in Jena University Hospital, aiming to include a representative range of severity levels (Sunnybrook severity scores: females 34 ± 14, males 27 ± 10). Three out of the twenty recruited patients were excluded at the beginning due to missing EMG data and unexcepted signal-to-noise ratio. Hence, the final study group consisted of 17 patients.

### 2.2. Experimental Setup and Paradigm

The bipolar EMG signals were captured with 4 mm Ag/AgCl cup-electrodes (EL254, Biopac Systems Inc., Goleta, CA USA), applied with electrode gel (NeurGel, GVB-geliMED GmbH, Bad Segeberg, Germany) and double-sided adhesive rings (GVB-geliMED GmbH, Germany). Measurements for the following extrinsic auricular muscles were acquired: anterior auricular muscle (AAM), superior auricular muscle (SAM), and posterior auricular muscle (PAM). Additionally, the EMG of the facial muscles zygomaticus major muscle (ZM) and orbicularis oculi muscle (OOM) on the contralateral (healthy) side were recorded and served later as a reference for data labeling. The reference electrode was a 30 mm × 22 mm Ag/AgCl self-adhesive, disposable electrode (Neuroline 720, Ambu GmbH, Theilheim, Germany) that was placed on the mastoid process of the contralateral side as a reference. All the electrodes were placed along the muscle fibers according to Figure 1. The skin at all electrode locations was cleaned with presaturated alcohol wipes and prepared with an abrasive gel (Nuprep, Weaver and Company, Tampa, FL USA) prior to application to reduce the electrode impedance. For measuring the EMG, amplifier FE238 Octal Bio Amp (AD Instruments Inc., Colorado Springs, CO, USA) in combination with the AD-Converter Powerlab 16/35 (AD Instruments Inc., USA; sampling rate: 10 kHz; analog high-pass filter at 0.1 Hz; digital notch filter at 50 Hz; automatically adjusted analog anti-aliasing filter) was utilized.

During the measurements, the patients were comfortably seated and performed different facial movements according to a paradigm video. The facial movements used are based on the standardized instruction video from the Facial Nerve Center Jena and are separated into target and interference movements [26]. Smile with lips closed (S), smile with teeth (ST), slight eye closure (SE), tight eye closure (TE), and eye blink (EB) were the target movements and were later be detected by the algorithm. The interference movements provided additional data for the algorithm, to be able to differentiate the target movements from other facial movements performed in daily life. These movements included frowning forehead (FF), wrinkled nose (WN), puckering lips (PL), clenched teeth (CT), mouth corners down (MC), open mouth (OM), and neutral position (NP). The paradigm itself comprises three distinct parts, each separated by a twenty-minute break. Each part comprises both target and interference movements, repeated consecutively ten times, lasting for three seconds and followed by three-second resting period. This was the same for all movements, except EB and NP. In case of EB, the movement was repeated 20 times with a provided pace of 30 blinks per minute. The higher repetition is due to the shorter time period compared to the other movements, which allows a more even distribution of windows per movement. The NP was performed at the beginning and at the end of each part for ten seconds. In each part, movements were presented in a different order to prevent bias. Additionally, everyday activities were included at the end of the paradigm (e.g., reading sentences, drinking, and chewing), and they were used to evaluate the electro-stimulation system in a real-life scenario. While conducting the measurements, the facial movements of the patients were captured with 60 frames per second using a video camera (Bio Stream, Logitech, Switzerland) to serve as an additional reference for labeling.

### 2.3. EMG Data Labeling

For the AI-based auricular-triggered algorithm, the EMG data of SAM, AAM, and PAM were labeled manually. The focus was on labeling the target movements—S, ST, SE, TE, and EB—based on a variety of movements. Only these target movements required accurate labeling from the start to the endpoint, as these are the movements that should be supported by the closed-loop electro-stimulation system. The labeling was performed manually using ELAN 6.6 (Eudico Linguistik Annotator, The Language Archive, Max Planck Institute for Psycholinguistics, Wundtlaan, The Netherlands) [27]. The EMG data recorded from the OOM and ZM of the contralateral side were synchronized with the video recording to enable accurate labeling by providing additional information through the video. Meanwhile, the auricular EMG data were hidden to prevent any bias. The facial movements EB, SE, and TE were labeled based on the EMG activity of the OOM of the contralateral side, because it shows electromyographic activity during these movements. The same was the case with the facial movements S and ST, which were labeled based on the EMG of the ZM of the contralateral side. All other facial movements were labeled as interference (IF). Parts of the data where the OOM or ZM signal displayed EMG activity that did not conform to the paradigm were excluded. For example, sections where the patient coughed or exhibited unintended movements were excluded from analysis.

### 2.4. Neural Network Training

The classification of the facial movements was performed using a Convolutional Recurrent Neural Network (CRNN) (Figure 2), which is the main part of the AI-based auricular-triggered algorithm. Due to significant variability in EMG activity among patients, stemming from the random nature of the synkinetic reinnervation process (as noted by Hochreiter et al. [18]), each patient was trained individually, which led to an individual ML model for each patient.

The labeled data were first downsampled from 10 kHz to 2 kHz. In surface EMGs, a cut-off frequency of around 450 Hz is standard, according to Kim et al. [28]. However, for auricular muscles, a maximum frequency of 1 kHz, which leads to a sampling rate of 2 kHz, was selected due to their minimal skin coverage, which lessens the tissues’ low-pass filtering effect. The labeled data were then normalized between 0 and 1 with the MinMaxScaler and the labels were encoded with the OneHotEncoder from the Scikit-Learn package to maximize the algorithm’s ability to learn effectively and efficiently. Subsequently, the data were divided into different datasets. Everyday activities were stored as a separate dataset, which was later used to test the electro-stimulation system in a more realistic setting. The other facial movements were split into training, validation, and test dataset. The first and second parts were split into approximately 80% training and 20% validation data. The last part of three in the paradigm was designed as test data. Each dataset contained an equal distribution of the different facial movements. Finally, each dataset was segmented into windows with a length of 66 ms, each having an overlap of 50% with adjacent windows. The chosen window length represents a compromise between capturing detailed features—in this case, the raw data within a single window—and minimizing the algorithm’s latency, which is important for a closed-loop electro-stimulation system performing in real time [28]. Following the segmentation, each window was assigned a label based on the most frequent label within it (Figure 3).

The CRNN configured with a batch size of 15 and a total of 60 epochs classified each EMG window as belonging to class S, ST, SE, TE, EB, or IF. The algorithm itself was trained using the training data. Throughout the training phase, the model’s performance was validated after each epoch against the validation data using the macro F1-score metric. The F1-score is a way to measure how good a model is at classification by looking at how many correct results it finds (accuracy) and how well it finds all the correct results (completeness). The macro F1-scrore provides an objective measure of the classifier’s performance, even with imbalanced data (Table 1), which occur due to the large number of combined interference movements. It averages the performance across classes, taking into account the total number of classes. It ranges from 0 (worst) to 1 (best) [29,30]. The model that achieved the highest validation macro F1-score was selected for the algorithm in the electro-stimulation system.

### 2.5. AI-Based Auricular-Triggered Algorithm

The electro-stimulation system contains an AI-based auricular-triggered algorithm that classifies the intended facial movement based on the extrinsic auricular muscle EMGs of the paretic side and generates a trigger for the stimulation of the corresponding facial muscle.

The input of this algorithm is a 66 ms window of EMG data from the three extrinsic auricular muscles. First, the MinMaxScaler needs to be applied to provide the same data quality as used during training of the model. The scaled EMG window is then classified by the CRNN, as explained in the previous section. It belongs to class S, ST, SE, TE, EB, or IF. Because the system should stimulate the according facial muscle during eye closure, eye blink, and smile, the classified movement needs to be categorized into the correct group. The facial movements S and ST were grouped into smile S, since both require a stimulation of the ZM. Similarly, the facial movements SE and TE were grouped into eye closure E, since both require a stimulation of the OOM. The third class is EB, which requires a shorter stimulation of the OOM, and the fourth is IF, which does not require any stimulation.

To avoid triggering stimulation based on a single EMG window, a stability feature is integrated at the end of the algorithm. Each target movement—S, E, and EB—has a corresponding counter. This counter increments when the same target movement is classified in consecutive windows. Once the counter reaches a maximum value of three for movements S and E, and two for movement EB, stimulation of the corresponding muscle is initiated. The stimulation then lasts for a predefined duration.

### 2.6. Closed-Loop Electro-Stimulation System

The electro-stimulation system should facilitate the communication between the extrinsic auricular muscles and the facial muscles, enabling communication between them in real time. A schematic of the system can be seen in Figure 4.

The process begins with capturing, amplifying, and filtering the EMG signals from the extrinsic auricular muscles on the paretic side (Figure 4: blue). Capturing was performed with the EMG amplifier FE232 Dual Bio Amp (AD Instruments Inc., Dunedin, New Zealand) with a sampling rate of 2 kHz, to align with the previously captured and processed data for the neural network. The amplifier integrates a digital 50 Hz notch filter and a digital 1 kHz low-pass filter of 1st order, which also serves as an anti-aliasing filter. An analog high-pass filter of 1st order with a cut-off frequency of 1 Hz was connected to the amplifier’s analog outputs to avoid low frequencies.

After amplification and filtering, the EMG signals are fed into a data acquisition (DAQ) system for signal processing, which generates the trigger for the stimulation of the ZM and OOM (Figure 4: green). This process was carried out using the DAQ system NI-USB 6002 from National Instruments (Austin, TX, USA), which is Python-compatible and communicates with the PC via a USB connection. The DAQ system, controlled by a Python script (Python 3.10.4 (Python Software Foundation, Wilmington, DE, USA)) with the AI-based auricular-triggered algorithm integrated, adjusts two analog outputs, which serve as triggers for the stimulation of the ZM and OOM.

To generate the stimulation waveform for the two distinct stimulations, the triggers of the ZM and OOM are sent to the Analog Discovery 2 function generator by Digilent Inc. (Pullman, USA) (Figure 4: red). This produces the voltage waveform upon activation of the stimulation. Finally, the voltage waveform is converted into a current waveform, which is facilitated by two STMISOLA surface stimulators from Biopac (Goleta, CA, USA) (Figure 3: yellow). The stimulators ensure the delivery of the appropriate stimulation current to the patient via stimulation electrodes. To further enhance patient safety, two switchboxes are integrated between the output of the STMISOLA and the stimulation electrode to enable quick disconnection of the patient from the system.

### 2.7. Evaluation of the Electro-Stimulation System

The evaluation of the system was conducted by means of a simulation. Instead of live input from the EMG amplifier, connected with the analog inputs of the DAQ system, the algorithm was directly fed with the already captured EMG data from the 17 patients. Therefore, the signals were downsampled from 10 kHz to 2 kHz to match the maximum frequency that would be used during live operation of the system.

The EMG signals were segmented into windows of appropriate lengths and processed concurrently to simulate real-time input. After completing a cycle, the system restarts with the next data segment without delay, unlike in real-time operation, in which it waits for sufficient samples. Consequently, the simulation does not reflect the time delay of the system. The stimulation duration for S and E was set to four seconds, because according to the paradigm, each movement was usually carried out for 3.97 s ± 0.64 s. The stimulation duration for EB was set to the mean duration for each patient separately.

The simulation was performed based on the test data and the everyday activities. In both cases, the trigger signal, which is the output of the DAQ system, was captured and visualized the stimulation of the different facial muscles. The evaluation was based on the macro F1-score. The true facial movements based on the labeled data were used as true values and the captured trigger signal as predicted value. To generate a representative amount of datapoints for the confusion matrix, both the signal representing the true facial movements and the trigger signal were sampled at a rate of 2 kHz. Out of these datapoints, the macro F1-score and the per-class F1-score were calculated. The result reflects the system’s behavior when deployed on patients.

### 2.8. Statistical Analysis

To analyze the performance differences across classes based on the per-class F1-score, the Kruskal–Wallis was used, followed by the Conover post hoc test, with an adjustment according to the Holm–Bonferroni method. This methodology was applied both to the test data and to data representing everyday activities, ensuring a comprehensive analysis.

Additionally, a correlation analysis, with the Spearman’s rank-order correlation coefficient ρ, was performed to identify any correlations between the macro F1-score of the algorithm based on the test data and various patient-related factors. These factors included age, sex, etiology, duration of the disease, and the severity of facial palsy and synkinesis. The null hypothesis suggested that there was no correlation between these variables. The correlation coefficient ρ varied between −1 and 1, with 0 implying no correlation [31]. This was undertaken for both the test data and the everyday activities.

## 3. Results

### 3.1. Simulated Performance of the Electro-Stimulation System Based on the Test Data

Figure 5 visualizes the simulated overall performance of the electro-stimulation system based on the test data, with a median macro F1-score of 0.602. Moreover, Figure 6 shows the per-class performance of the system for the test data. According to the Kruskal–Wallis test followed by the Conover post hoc test, significant differences between most of the classes with *p*-values less than 0.05 were observed (see Appendix A). However, there was no significant difference between the class smile/smile with teeth and the class slight/tight eye closure (*p* = 0.951).

### 3.2. Simulated Performance of the Electro-Stimulation System Based on the Everyday Activities

Figure 7 displays the overall performance of the electro-stimulation system, with a median macro F1-score of 0.274. In addition, Figure 8 depicts the per-class performance of the system on everyday activities. The Kruskal–Wallis test and subsequent Conover post hoc test revealed significant differences with *p*-values less than 0.05 between most of the classes (see Appendix A). Nonetheless, the smile/smile with teeth and eye blink classes did not show a significant difference (*p* = 906).

### 3.3. Correlation Analysis between the Macro F1-Score and Any Other Patient-Related Factor

The correlation analysis revealed a correlation between the macro F1-score based on the test data and the severity of synkinesis (based on the Sunnybrook score) of ρ = 0.570. This correlation was statistically significant (*p* = 0.017). No other correlations were found between the macro F1-score and other patient-related factors.

## 4. Discussion

The present study demonstrated that extrinsic auricular muscles, due to their specific activity patterns during various facial movements, can be used to develop an electro-stimulation system with a ML algorithm, classifying facial movements based on the EMG of the extrinsic auricular muscles and stimulating the according facial muscles.

The results depicted in Figure 5, Figure 6, Figure 7 and Figure 8 demonstrate significant differences in the classification performance across the different patients. Several factors could have contributed to this variability. Differences between the patients in aspects such as age, gender, muscle tone, and variations in skin conductivity could have resulted in this variability, even though no correlations could be found between most of the patient specific parameters. While care was taken to standardize the placement of the electrodes, minor deviations in placement or impedance between skin and electrode could have had a significant impact on signal quality and, therefore, on the overall performance. The same was the case for the external noise and interference. Moreover, different facial movements might have been performed different by each patient, in terms of both intensity and consistency. Additionally, the severity of facial palsy and the presence of synkinesis have a significant impact on the system’s performance. Notably, the higher the severity of the synkinesis (based on the Sunnybrook score), the better the performance of the electro-stimulation system. This confirms the assumption that synkinetic reinnervation provides more information in auricular muscle EMGs regarding facial movements, leading to a better trigger signal for the electro-stimulation system. These findings correlate with those reported by Hochreiter et al. [18]. However, the obtained macro F1-scores are still insufficient for the development of a viable product.

Table 1 summarizes the correct, incorrect, and missed stimulations for the median (macro F1-score = 0.632) and the best (macro F1-score = 0.782)-performing patient. It is evident that most movements, except for eye blinking, were supported by correct stimulation. Nevertheless, many incorrect stimulations were present for both patients.

According to Table 2 and Figure 6, it seems like the system performs worse in the classification of eye blink compared to slight/tight eye closure. A statistically significant difference was proven between the two (*p* = 0.003). However, this finding shall be interpreted with caution, because eye blink is the minority class, which can automatically lead to a worse score. Incorrect stimulations intended to elicit an eye blink commonly occur at the beginning of slight/tight eye closure, because the algorithm occasionally cannot differentiate between an eye closure and an eye blink. However, both classified movements ultimately lead to stimulation of the same target muscle. As can be seen in Figure 6, slight/tight eye closure shows no statistically significant difference compared to smile/smile with teeth (*p* = 0.578). However, the interquartile range (IQR) of smile/smile with teeth is much smaller compared to the eyelid movements, which could indicate that eyelid movements are more difficult to classify. This is in line with the findings of Lipede et al. [32]. They have shown that PAM activity increased significantly from baseline during smiling compared to other facial movements. Additionally, false-positive classifications greatly reduce the performance of the system. For example, incorrect stimulations intended to elicit a smile/smile with teeth are mostly due to a lack of differentiation between smiling and clenched teeth. Furthermore, more subtle movements, like slight eye closure or smiling with the lips closed, performed much worse for most of the patients than tight eye closure or smiling with teeth. This seems reasonable, as it can be assumed that more subtle facial movements involve less activity in the facial muscles and in the auricular muscles. The study by Lipede et al. underlines this finding [32].

Compared to the test data, the performance of the electro-stimulation system during everyday activities was much worse. This led to a macro F1-score of 0.291 for the median-performing patient and to a macro F1-score of 0.350 for the best-performing patient according to the test data. The EMG patterns and the stimulations during everyday activities are random and have nothing to do with the performed facial movement. For example, when chewing gum, extremely high auricular EMG activities are observed, which could be due to cross-talk. Rantanen et al. noted that high cross-talk affects all facial muscles during chewing [15]. Although auricular-muscle EMGs were used in the present study, it is obvious that cross-talk also impacts these EMGs, because the different chewing muscles lie anterior and superior to the auricle [33]. Therefore, the present electro-stimulation system, with this algorithm, cannot be used during everyday activities.

### 4.1. Comparison with Other Studies

Compared to other studies focused on an electro-stimulation system for patients with facial palsy, the present study is the only one supporting the three facial movements eye blink, eye closure, and smile [14,16]. Previous studies focused solely on detecting eye blinks. The present study uses the extrinsic auricular EMG signals from the paretic side, preserving the healthy side of the patient, which leads to its main advantage in comparison to other studies. Previous studies used the facial muscle EMG signals of the healthy side. Due to the fundamental differences in setup and methods between the different studies, especially given that all the other studies focused only on the eye blink, comparisons of the performances cannot be performed.

### 4.2. Limitations and Strengths

One significant limitation of the present study is the relatively small sample size, which may limit the generalizability of the findings to a broader patient population. A larger sample size would provide more robust data and a better understanding of the model’s performance across patients. The developed system is only applicable to synkinetic reinnervated patients and not to deinnervated patients, which restricts its applicability to a subset of individuals suffering from facial palsy. In addition, the preparation of the model requires a considerable amount of time, particularly in terms of data collection, labeling and training, which could limit the practicality of the system in real-world clinical settings. Furthermore, the system’s use is restricted to predefined facial movements—specifically eye blinking, eye closure, and smiling—and does not extend to everyday activities, which makes the system applicable only in rehabilitation training.

Despite these limitations, the present study offers several strengths compared to existing research in the field. The system supports three distinct facial movements—eye blink, eye closure, and smile—which provides a higher degree of support compared to already existing studies. Another key strength is the use of the paretic side as a trigger source, which leads to the preservation of the healthy side of the patient and complexity reduction for the system.

### 4.3. Outlook

In order to use such a system, the algorithm still requires improvement to eliminate incorrect stimulations, especially in accurately classifying similar movements such as smiling with teeth and clenched teeth. The needed improvements in performance could be achieved by implementing another ML algorithm. For instance, a more robust recurrent neural network with stronger sequence dependencies might reduce fluctuations between predictions, resulting in better performance. To gain a better performance during everyday activities, the training dataset needs to be reconsidered. Integrating data from everyday activities into the training set could be a strategy to achieve better scores. The algorithm must adjust to poorer data quality due to the appearance of various everyday activities and cross-talk. Collecting auricular EMG signals during daily life and using them as training data could be another possibility. This would lead to more realistic training data, thereby improving the system’s performance.

In a future step, the electro-stimulation system needs to be tested live on patients, as it has only been evaluated offline through simulation so far. This will enable the evaluation of the system’s real-time performance and provide further proof of its functionality and side-effects. Another question that needs to be addressed in the future is the reproducibility of the electrode positions to ensure similar EMG signals for the good performance of the ML algorithm. However, this study makes valuable contributions toward the development of an electro-stimulation system for facial muscle stimulation in peripheral facial palsy.

## 5. Conclusions

The present study demonstrated that the extrinsic auricular EMG signals from the synkinetic side contain information that can be used as a trigger for an electro-stimulation system for patients with facial palsy. However, the algorithm still requires improvement, especially in accurately classifying similar movements, such as smiling with teeth and clenched teeth. In summary, the study showed that an electro-stimulation system with an AI-based auricular-triggered algorithm can be used to some extent to support facial movements on the paretic side in patients with peripheral facial palsy. However, the usability of the present system is limited to predefined movements, making it suitable to some extent for use in rehabilitation training. It is not suitable for use in daily life because reliable classification is not possible due to high cross-talk, which primarily occurs because of speaking and chewing.

## Figures and Tables

**Figure 1 diagnostics-14-02158-f001:**
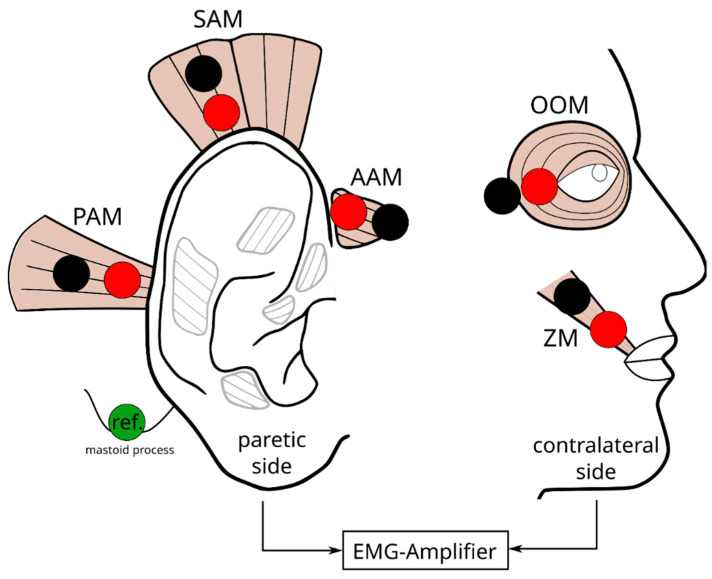
Schematic of the experimental setup. The red and black dots visualize the electrodes (cathode and anode). The green dot represents the reference electrode. Anterior auricular muscle (AAM), superior auricular muscle (SAM), posterior auricular muscle (PAM), zygomaticus major muscle (ZM), and orbicularis oculi muscle (OOM).

**Figure 2 diagnostics-14-02158-f002:**
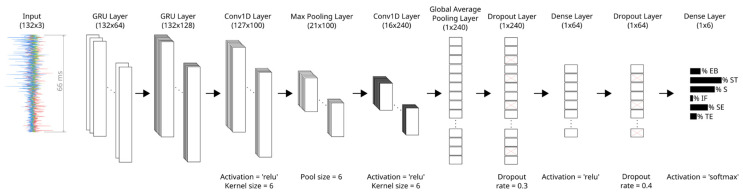
CRNN with different layers and parameters. The input data represent one EMG window for one patient during the movement ST (132 × 3 = number of samples in one window x number of EMG channels). Each layer has a dimensionality that is defined by the number of samples in one window x number of units. The final layer is a dense layer with an output size of six, corresponding to the six different classes. The predicted label is ST. The model was trained with a batch size of 15 and 60 epochs. The three colors in the EMG image on the left side illustrate the three different auricular muscle recordings.

**Figure 3 diagnostics-14-02158-f003:**
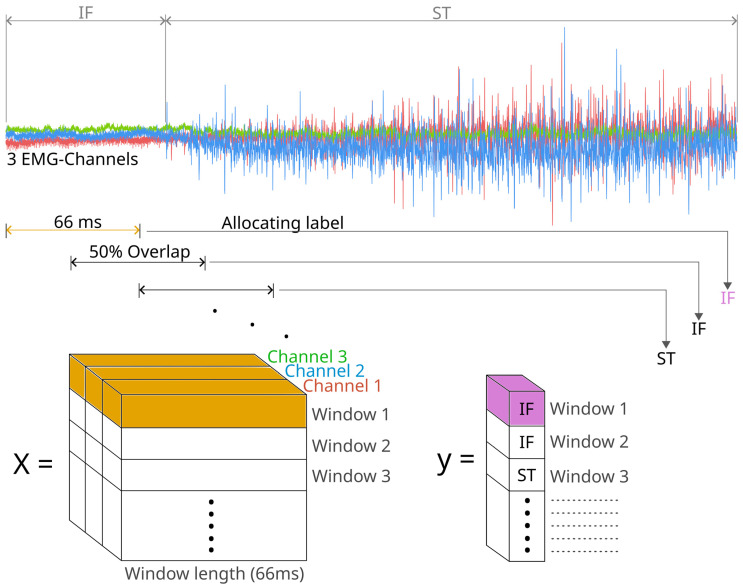
Visualization of the window-feature allocation based on the three labeled extrinsic auricular EMG signals. One EMG window with a length of 66 ms (yellow) (containing 132 samples per channel = 132 × 3 = 396) visualizes one input of the Neural Network. Each window receives one label (purple), depending on the most frequent label within it. All the different windows with all the datapoints are saved into the matrix X and all the according labels into the vector y. The three colors in the EMG image in the upper part of the figure illustrate the three different auricular muscle recordings.

**Figure 4 diagnostics-14-02158-f004:**
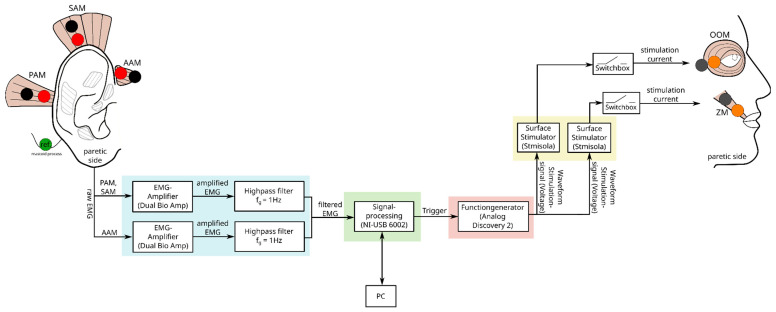
Schematic of the electro-stimulation system for patients with peripheral facial palsy. The red and black dots represent the EMG electrodes and the orange and gray dots the stimulation electrodes. The green dot visualizes the reference electrode.

**Figure 5 diagnostics-14-02158-f005:**
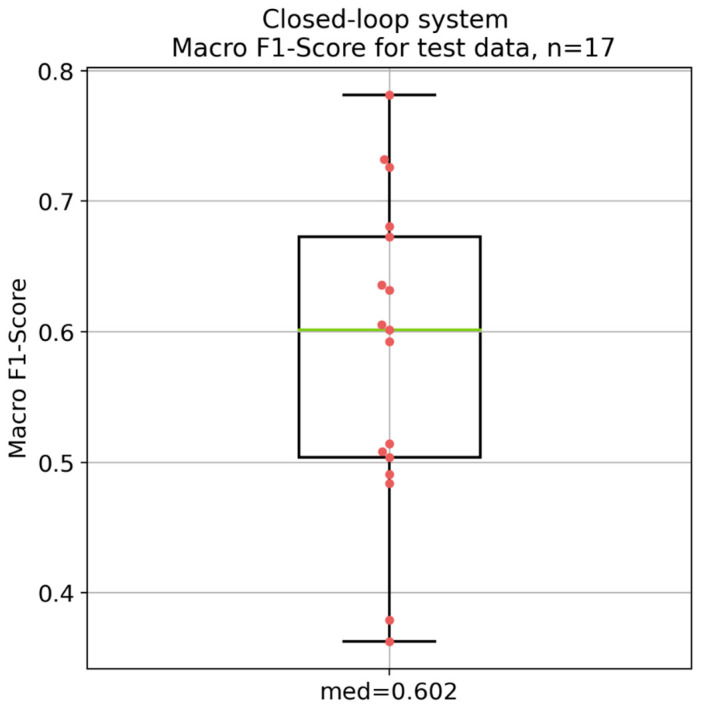
Overall performance of the electro-stimulation system based on the macro F1-score for the test data from the 17 patients. *n* = number of patients; med = median macro F1-score. Green: median value; red dots: individual values of each patient; black lines: 25% and 75% quartile.

**Figure 6 diagnostics-14-02158-f006:**
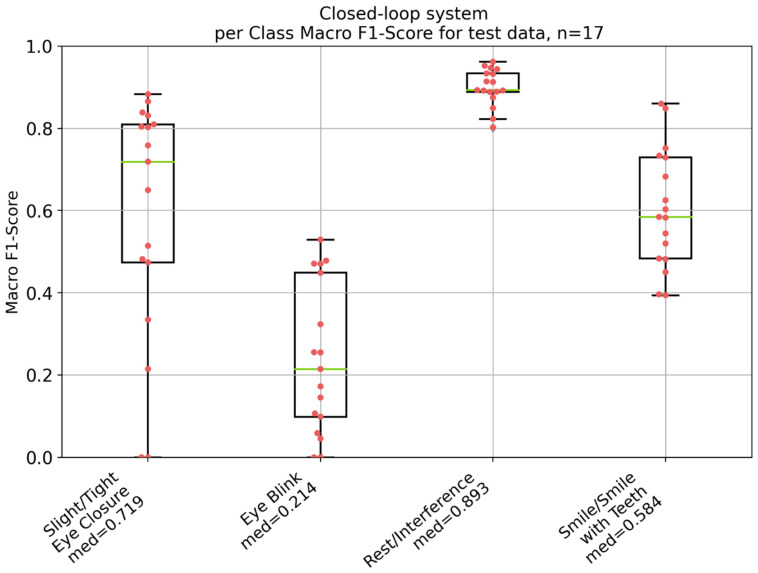
Per-class performance of the electro-stimulation system based on the per-class F1-score for the test data of the 17 patients. *n* = number of patients; med = median macro F1-score. Green: median value; red dots: individual values of each patient; black lines: 25% and 75% quartile.

**Figure 7 diagnostics-14-02158-f007:**
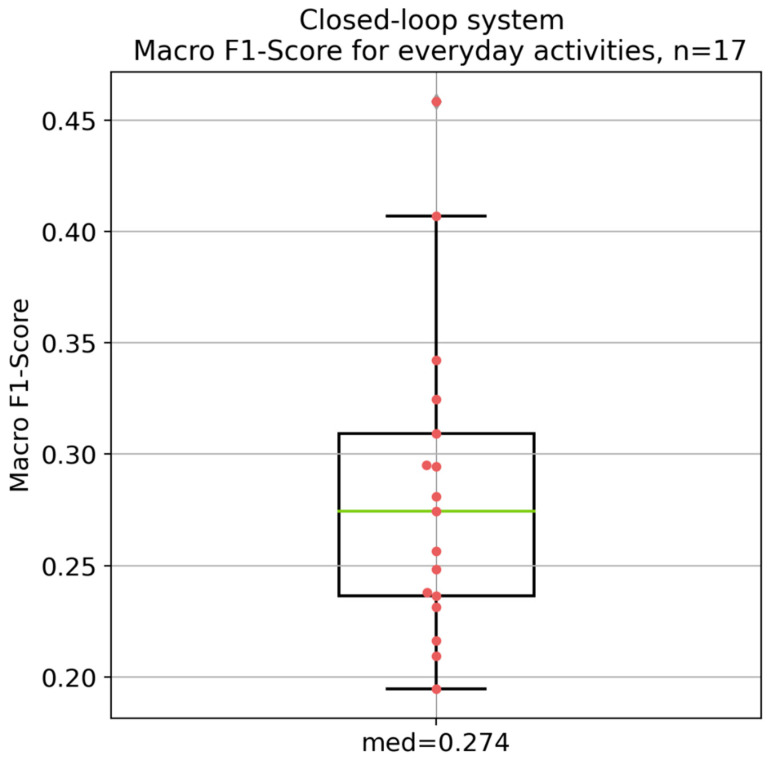
Overall performance of the electro-stimulation system based on the macro F1-score for the everyday activities of the 16 patients. *n* = number of patients; med = median macro F1-score. Green: median value; red dots: individual values of each patient; black lines: 25% and 75% quartile.

**Figure 8 diagnostics-14-02158-f008:**
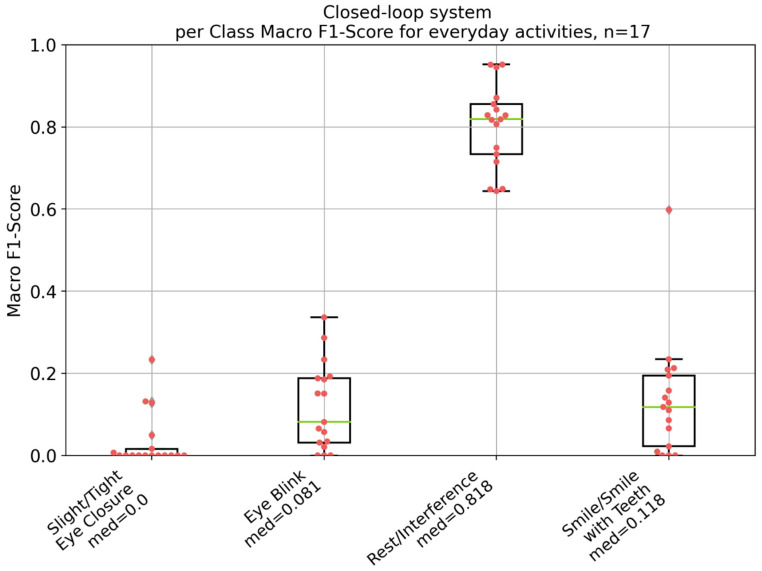
Per-class performance of the electro-stimulation system based on the per-class F1-score for the everyday activities of the 17 patients. *n* = number of patients; med = median macro F1-score. Green: median value; red dots: individual values of each patient; black lines: 25% and 75% quartile.

**Table 1 diagnostics-14-02158-t001:** Absolute amount of data available per class. The value visualizes the number of windows with a length of 66 ms, containing the values of three different channels. The value 1, for example, represents the yellow highlighted area from the matrix X, in Figure 3. The total number of data was calculated from windows with a length of 66 ms and 50% overlap.

	Training, Validation, and Test Data					
	S	ST	SE	TE	EB	IF
Mean	3284.1	3328.5	3413.2	3500.9	1442.6	69,640.1
Standard deviation	277.2	387.5	198	199.7	1455	8105.9
Minimum	2640	2307	3090	3157	255	54,929
Maximum	3649	3962	3759	3816	6666	82,242
	Everyday activities					
	S	ST	SE	TE	EB	IF
Mean	342.2	390.5	66.6	2.4	415.3	16,563.7
Standard deviation	315.5	397.9	110.6	9.7	27.8	2336.5
Minimum	0	0	0	0	68	11,180
Maximum	1101	1499	415	40	1147	18,780

Abbreviations: Smile with lips closed (S), smile with teeth (ST), slight eye closure (SE), tight eye closure (TE), eye blink (EB); interference (IF).

**Table 2 diagnostics-14-02158-t002:** System’s performance for the median and best-performing patient.

Median-Performing Patient (Patient 5)				
	True Movement	Correct Stim.	Incorrect Stim.	Missed Stim.
smile/smile with teeth	20	18	9	2
slight/tight eye closure	20	19	3	1
eye blink	20	6	18	13
best performing patient (patient 2)				
	true movement	correct stim.	incorrect stim.	missed stim.
smile/smile with teeth	20	20	3	0
slight/tight eye closure	20	20	1	0
eye blink	66	63	12	7

True mov. is based on the number of performed movements. Correct stim.: number of stimulations of the correct target muscle, incorrect stim.: number of stimulations of the wrong target muscle, missed stim.: failed to stimulate a target muscle when it should have.

## Data Availability

The datasets used during the current study are available from the corresponding author upon reasonable request.

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
