# Peer review of "Electro-Stimulation System with Artificial-Intelligence-Based Auricular-Triggered Algorithm to Support Facial Movements in Peripheral Facial Palsy: A Simulation Pilot Study"

_diagnostics, 2024, doi:10.3390/diagnostics14192158_

Round 1

Reviewer 1 Report

Comments and Suggestions for Authors

The authors proposed a study focused on developing a closed-loop electro-stimulation system to help patients with facial palsy. In the proposed method, the system detects the movements, it provides targeted electrical stimulation to the appropriate facial muscles on the affected side, aiming to mimic normal facial expressions. This system, based on artificial intelligence and triggered by signals from the auricular muscles, represents a significant advancement in treatment. By delivering targeted electro-stimulation, the proposed system could greatly improve the daily lives of facial palsy patients.

The paper is well-written and organized. The details are given for readers to understand the paper better. I have some questions.

 1-      There are 17 patients. How many signals did you take from patients? What is the number of samples for each class?

2-      It would be better to add a confusion matrix for the classification result of CRNN. If the authors think Table 1 is enough to show input and output results, I leave the decision to authors about adding the confusion matrix for CRNN.  

3-       What is the response time of your system? Is it measurable?

Author Response

Rebuttal letter

Manuscript ID: diagnostics-3195862

Title: Electro-Stimulation System with Artificial-Intelligence-Based Auricular-Triggered Algorithm to Support Facial Movements in Peripheral Facial Palsy: A Simulation Pilot Study

Authors: Katharina Steiner, Marius Taro Arnz, Gerd Fabian Volk, Orlando Guntinas-Lichius *

We like to thank the reviewers for the helpful comments. We would like to answer all queries point-by-point.

Reviewer 1

The authors proposed a study focused on developing a closed-loop electro-stimulation system to help patients with facial palsy. In the proposed method, the system detects the movements, it provides targeted electrical stimulation to the appropriate facial muscles on the affected side, aiming to mimic normal facial expressions. This system, based on artificial intelligence and triggered by signals from the auricular muscles, represents a significant advancement in treatment. By delivering targeted electro-stimulation, the proposed system could greatly improve the daily lives of facial palsy patients.

The paper is well-written and organized. The details are given for readers to understand the paper better. I have some questions.

1.1. There are 17 patients. How many signals did you take from patients? What is the number of samples for each class?

Answer 1.1:

Thank you for these questions. The number of signals is already mentioned in Section 2.2:

“The following extrinsic auricular muscles were acquired: anterior auricular muscle (AAM), superior auricular muscle (SAM), and posterior auricular muscle (PAM).”

The number of samples for each class was not mentioned directly, but we have now added it to Section 2.4, Table 1. Visualizing the amount of data used in this study is indeed a good idea.

1.2. It would be better to add a confusion matrix for the classification result of CRNN. If the authors think Table 1 is enough to show input and output results, I leave the decision to authors about adding the confusion matrix for CRNN.  

Answer 1.2:

Thank you for the suggestion. We had been considering whether to use a table or a confusion matrix to visualize the input and output results. We ultimately decided to use a table because it presents the results in a simpler and more abstract way. The confusion matrix would only provide additional information on the stimulated area (eliciting a smile, eye closure, or eye blink) for incorrect stimulations, but that information is unnecessary in this context.

1.3. What is the response time of your system? Is it measurable?

Answer 1.3:

The response time would be measurable if the system is directly applied on the patient. Nevertheless, the present paper deals with the simulation of the system, which provides no time information about the system. The response time needs to be evaluated in a future step, as it is mentioned in the discussion:

“In a future step, the electro-stimulation system needs to be tested live on patients, as it has only been evaluated offline through simulation so far. Doing so will enable evaluation of the systems real-time performance and provide further proof of its functionality and side-effects.”

Katharina Steiner and Orlando Guntinas-Lichius

for all authors

Innsbruck and Jena, 25-September-2024

Reviewer 2 Report

Comments and Suggestions for Authors

This study developed an AI-based closed-loop electrical stimulation system that used ear EMG signals to detect facial movements and targeted stimulation of paralyzed side muscles to help restore smile and eyelid closure functions. Through testing on 17 patients, the system shows good results (F1 score 0.602), providing a non-invasive and intelligent personalized treatment scheme for patients with facial paralysis.

1. By clearly articulating how your work addresses the limitations of existing approaches and illustrating its unique contribution, you will give the reader a better understanding of how your research fits into the field.

2. Visualizing the dataset greatly enhances the reader's intuitive understanding of your work. By presenting the data distribution, characteristics, quantities, and processes as graphs or images, you can quickly grasp the most important features of the data.

3. Richer classification evaluation metrics (e.g., accuracy, precision, recall) can help evaluate model performance more comprehensively.

4. The lack of comparative experiments with current state-of-the-art methods in the manuscript limits a comprehensive evaluation of the actual performance of the proposed model.

5. In the experimental results in Figures 4, 5, 6, and 7, there are significant differences in the classification results among different patients, and it is necessary to analyze the possible reasons for these differences.

Author Response

Manuscript ID: diagnostics-3195862

Title: Electro-Stimulation System with Artificial-Intelligence-Based Auricular-Triggered Algorithm to Support Facial Movements in Peripheral Facial Palsy: A Simulation Pilot Study

Authors: Katharina Steiner, Marius Taro Arnz, Gerd Fabian Volk, Orlando Guntinas-Lichius *

We like to thank the reviewers for the helpful comments. We would like to answer all queries point-by-point.

Reviewer 2

This study developed an AI-based closed-loop electrical stimulation system that used ear EMG signals to detect facial movements and targeted stimulation of paralyzed side muscles to help restore smile and eyelid closure functions. Through testing on 17 patients, the system shows good results (F1 score 0.602), providing a non-invasive and intelligent personalized treatment scheme for patients with facial paralysis.

2.1. By clearly articulating how your work addresses the limitations of existing approaches and illustrating its unique contribution, you will give the reader a better understanding of how your research fits into the field.

Answer 2.1:

Thank you for the advice. To articulate the limitations of existing approaches and the objective of our work more clearly, we have revised the introduction of the paper.

2.2. Visualizing the dataset greatly enhances the reader's intuitive understanding of your work. By presenting the data distribution, characteristics, quantities, and processes as graphs or images, you can quickly grasp the most important features of the data.

Answer 2.2:

Thank you for these ideas. A similar question was already mentioned by reviewer 1. we added the number of samples (incl. mean, standard deviation, minimum and maximum) of each class in Section 2.4, Table 1. It gives more details about the amount of data and their distribution. Additionally, I added Figure 3 in Section 2.4, to get a better understanding about how the input data of the neural network is configured. There are much more possibilities to visualize the data and all the different processes which were performed. Although we would not add more figures and tables because all other processes are explained in detail in the text. Additionally, more information regarding the dataset itself is not necessary because the neural network is fed with the data visualized in Figure 3. Even though, If the reviewer has precise ideas about how to visualize the dataset or the processes, we would appreciate that.

2.3. Richer classification evaluation metrics (e.g., accuracy, precision, recall) can help evaluate model performance more comprehensively.

Answer 2.3:

Of course, richer metrics can provide deeper insights into the model's performance. However, since this is a first pilot study and the dataset is imbalanced, we deliberately chose the F1-macro score. Using accuracy would result in a higher score for the algorithm, but this would be misleading as it would nott accurately reflect its true performance. The F1-score balances precision and recall equally, which is why we decided not to mention precision and recall separately. Therefore, we chose not to use additional metrics to evaluate the model.

2.4. The lack of comparative experiments with current state-of-the-art methods in the manuscript limits a comprehensive evaluation of the actual performance of the proposed model.

Answer 2.4:

                We already had written a short paragraph which compares the present study with similar ones:

“Compared to other studies focused on an electro-stimulation system for patients with facial palsy, the present one is the only one supporting the three facial movements eye blink, eye closure and smile [14] [33]. Already existing studies focused solely on detecting eye blinks. The present study uses the extrinsic auricular EMG signals from the paretic side, preserving the healthy side of the patient, which leads to the main advantage in comparison to other studies. Already existing studies used the facial muscle EMG signals of the healthy side. Due to the fundamental differences in setup and methods between the different studies, especially given that all other studies focused only on the eye blink, comparisons of the performance can’t be performed.”

However, this comparison should be treated with caution due to differences in setup and methods. A direct comparison of performance is not possible, as the other studies focus solely on detecting eye blinks. Cervera-Negueruela et al. achieved an accuracy of 95% for detecting eye blinks. However, due to differences in the paradigm and the imbalanced dataset in the present study, the accuracy cannot be used to compare the performance of both studies. Nonetheless, the present study offers two advantages over others: it supports eye blink, eye closure and smiling, and it uses the paretic side, preserving the healthy side.

2.5. In the experimental results in Figures 4, 5, 6, and 7, there are significant differences in the classification results among different patients, and it is necessary to analyze the possible reasons for these differences.

Answer 2.5:

Thank you for this advice. I added the following paragraph into section 4 “Discussion”:

“The results depicted in Figure 4 – 7 demonstrate significant differences in the classification performance across the different patients. Several factors could contribute to this variability. Differences between the patients such as age, gender, muscle tone and variations in skin conductivity could result in this variability, even though no correlation could be found between most of the patient specific parameters. While care was taken to standardize the placement of electrodes, minor deviations in placement or impedance between skin and electrode could have a significant impact on signal quality and therefore on the overall performance. The same with external noise and interference. Moreover, the facial movements might be performed different by each patient, both in terms of intensity and consistency. Additionally, the severity of facial palsy and the presence of synkinesis have a significant impact on the system’s performance. Notably, the higher the severity of synkinesis (based on the Sunnybrook score), the better the performance of the electro-stimulation system. It confirms the assumption that synkinetic reinnervation provides more information in the auricular muscle EMGs regarding facial movements, leading to a better trigger signal for the electro-stimulation system. These findings correlate with those reported by Hochreiter et al. [16]. However, the obtained macro F1-scores are still insufficient for the development of a viable product.“

Katharina Steiner and Orlando Guntinas-Lichius

for all authors

Innsbruck and Jena, 25-September-2024

Reviewer 3 Report

Comments and Suggestions for Authors

The purpose of this research is to investigate a closed-loop electro-stimulation system as treatment for the facial palsy. The study showed that such this system, using an AI-based auricular-triggered algorithm, can support the facial movements on the synkinetic side in patients with unilateral chronic facial palsy with synkinesis.

The study is potentially interesting, the methods are sounding to me, the work seems well performed and sufficiently detailed, the figures and tables are clear.
Some points remain to be clarified:

1)In study population the authors write “Twenty patients (10 female, 55±37 years; 7 male, 62±11 years”, but 10 female and 7 male are 17 patients. There is obviously a mistake. Even though three patients were excluded I would suggest adding the characteristics of the three excluded subjects.

2)It is not clear how eye blinks, eye closure and smile were measured. It is important to specify, since they were used for correlations.

3)It would be useful to add a paragraph with limits and strengths. The small sample size should definitely be included among the limits.

Author Response

Rebuttal letter

Manuscript ID: diagnostics-3195862

Title: Electro-Stimulation System with Artificial-Intelligence-Based Auricular-Triggered Algorithm to Support Facial Movements in Peripheral Facial Palsy: A Simulation Pilot Study

Authors: Katharina Steiner, Marius Taro Arnz, Gerd Fabian Volk, Orlando Guntinas-Lichius *

We like to thank the reviewers for the helpful comments. We would like to answer all queries point-by-point.

Reviewer 3

The purpose of this research is to investigate a closed-loop electro-stimulation system as treatment for the facial palsy. The study showed that such this system, using an AI-based auricular-triggered algorithm, can support the facial movements on the synkinetic side in patients with unilateral chronic facial palsy with synkinesis.

The study is potentially interesting, the methods are sounding to me, the work seems well performed and sufficiently detailed, the figures and tables are clear.

Some points remain to be clarified:

3.1. In study population the authors write “Twenty patients (10 female, 55±37 years; 7 male, 62±11 years”, but 10 female and 7 male are 17 patients. There is obviously a mistake. Even though three patients were excluded I would suggest adding the characteristics of the three excluded subjects.

Answer 3.1:

That was a mistake from our side. We regret this. We added the characteristics of the three excluded patients.

“Twenty patients (12 female, 49±13 years; 8 male, 58±15 years) diagnosed with unilateral peripheral chronic (facial palsy duration: female 27±15 months, male 77 ±105 months) facial palsy with synkinesis (Sunnybrook [23] synkinesis score: female 9±2, male 9±3) were recruited. All patients were selected from the Facial-Nerve-Center in Jena University Hospital aiming to include a representative range of severity levels (Sunnybrook severity score: female 34±14, male 27±10). Three out of the twenty recruited patients were excluded at the beginning due to missing EMG data and unexcepted signal-to-noise ratio. Hence, the final study group consisted of 17 patients.“

3.2. It is not clear how eye blinks, eye closure and smile were measured. It is important to specify, since they were used for correlations.

Answer 3.2:

We mentioned in Section 2.2 the different signals Ie measured:

“The following extrinsic auricular muscles were acquired: anterior auricular muscle (AAM), superior auricular muscle (SAM), and posterior auricular muscle (PAM). Additionally, the EMG of the facial muscles zygomaticus major muscle (ZM) and orbicularis oculi muscle (OOM) from the contralateral (healthy) side were recorded and serve later as a reference for data labeling.”

“While conducting the measurements the facial movements of the patients were captured with 60 frames per second using a video camera (Bio Stream, Logitech, Switzerland) to serve as an additional reference for labeling.”

Additionally we adapted the Section 2.3 “EMG Data Labeling” to explain more precisely how we got from the raw signals to the labeled signals:

“For the AI-based auricular-triggered algorithm, the EMG data of SAM, AAM and PAM were labeled manually. The focus was on labeling the target movements – S, ST, SE, TE and EB – from a variety of movements. Only these target movements required accurate labeling from the start to the endpoint, as these are the movements that should be supported by the closed-loop electro-stimulation system. The labeling was done manually using ELAN 6.6 (Eudico Linguistik Annotator, The Language Archive, Max Planck Institute for Psycholinguistics, Netherlands) [25]. The EMG data recorded from the OOM and ZM of the contralateral side were synchronized with the video recording to enable accurate labeling by providing additional information through the video. Meanwhile, the auricular EMG data was hidden to prevent any bias. The facial movements EB, SE and TE were labeled based on the EMG activity of the OOM of the contralateral side, because it shows electromyographic activity during these movements. The same with the facial movements S and ST which were labeled based on the EMG of the ZM of the contralateral side. All other facial movements were labeled as interference (IF). Parts of the data where the OOM or ZM signal displayed EMG activity that did not conform to the paradigm were excluded. For example, sections where the patient coughed or exhibited unintended movements were excluded from analysis. “

If anything is unclear or if we have misunderstood the reviewer’s comment, please be so kind to provide more detailed information.

3.3. It would be useful to add a paragraph with limits and strengths. The small sample size should definitely be included among the limits.

Answer 3.3:

Thank you for this advice. For this purpose, we have split the Chapter “Discussion” into different Sections. I added the following content into Section 4.2. “Limitations and Strengths”:

“One significant limitation of the present study is the relatively small sample size, which may limit the generalizability of the findings to a broader patient population. A larger sample size would provide more robust data and a better understanding of the model’s performance across patients. The developed system is only applicable to synki-netic reinnervated patients and not to deinnervated patients, which restricts its applicabil-ity to a subset of individuals suffering from facial palsy. In addition, the preparation of the model requires a considerable amount of time, particularly in terms of data collection, la-beling and training, which could limit the practicality of the system in real-world clinical settings. Furthermore, the system's use is restricted to predefined facial move-ments—specifically eye blink, eye closure, and smiling—and does not extend to everyday activities, which makes the system applicable only in rehabilitation training.

Despite these limitations, the present study offers several strengths compared to ex-isting research in the field. The system supports three distinct facial movements – eye blink, eye closure and smile – which provides a higher degree of support compared to al-ready existing studies. Another key strength is the use of the paretic side as a trigger source, which leads to the preservation of the healthy side of the patient and a complexity reduction of the system.”

Katharina Steiner and Orlando Guntinas-Lichius

for all authors

Innsbruck and Jena, 25-September-2024

Round 2

Reviewer 2 Report

Comments and Suggestions for Authors

All my concerns have been addressed.

Reviewer 3 Report

Comments and Suggestions for Authors

The authors responded satisfactorily, so the manuscript can be published in its current form.